# Beyond Genetics: Metastasis as an Adaptive Response in Breast Cancer

**DOI:** 10.3390/ijms23116271

**Published:** 2022-06-03

**Authors:** Federica Ruscitto, Niccolò Roda, Chiara Priami, Enrica Migliaccio, Pier Giuseppe Pelicci

**Affiliations:** 1European Institute of Oncology (IEO) IRCCS, Via Ripamonti 435, 20141 Milan, Italy; federica.ruscitto@ieo.it (F.R.); niccolo.roda@ieo.it (N.R.); chiara.priami@ieo.it (C.P.); 2Department of Oncology and Hemato-Oncology, University of Milan, Via Santa Sofia 9, 20142 Milan, Italy

**Keywords:** breast cancer, metastatic cascade, intra-tumor heterogeneity, mutational profile, adaptive responses

## Abstract

Metastatic disease represents the primary cause of breast cancer (BC) mortality, yet it is still one of the most enigmatic processes in the biology of this tumor. Metastatic progression includes distinct phases: invasion, intravasation, hematogenous dissemination, extravasation and seeding at distant sites, micro-metastasis formation and metastatic outgrowth. Whole-genome sequencing analyses of primary BC and metastases revealed that BC metastatization is a non-genetically selected trait, rather the result of transcriptional and metabolic adaptation to the unfavorable microenvironmental conditions which cancer cells are exposed to (e.g., hypoxia, low nutrients, endoplasmic reticulum stress and chemotherapy administration). In this regard, the latest multi-omics analyses unveiled intra-tumor phenotypic heterogeneity, which determines the polyclonal nature of breast tumors and constitutes a challenge for clinicians, correlating with patient poor prognosis. The present work reviews BC classification and epidemiology, focusing on the impact of metastatic disease on patient prognosis and survival, while describing general principles and current in vitro/in vivo models of the BC metastatic cascade. The authors address here both genetic and phenotypic intrinsic heterogeneity of breast tumors, reporting the latest studies that support the role of the latter in metastatic spreading. Finally, the review illustrates the mechanisms underlying adaptive stress responses during BC metastatic progression.

## 1. Breast Cancer Mortality Is Associated with Metastatic Disease

Breast cancer (BC) arises from the transformation of epithelial cells of the ductal-lobular compartment of the mammary gland [1] and it accounts for ~30% of diagnosed cancers and ~15% of cancer-related deaths in women [2]. BC incidence increases with age, being maximal between 50–70 years [3] and it is tightly linked to ethnicity, with African American women displaying the highest incidence and worst prognosis [4,5]. Several risk factors are associated with BC [6], including a family history of BC, due to inherited variants of cancer predisposing genes, such as BRCA1 and BRCA2 [7], early menarche and late menopause [8], obesity [9,10], alcohol consumption [11], physical inactivity [12] and exposure to exogenous hormones (e.g., oral contraceptives and menopausal hormone replacement therapy, [13]).

Molecular classification [14] stratifies BC patients into four major groups [15] on the basis of the expression of estrogen receptor (ESR), progesterone receptor (PR), human epidermal growth factor 2 receptor (HER2) and the proliferative marker Ki67. Tumors classified as Luminal A and B express both ESR and PR, with the A subtype displaying higher expression levels and B tumors occasionally expressing also HER2. The proliferation rate in luminal tumors is variable, but it is generally higher in the B subtype. Consistently, prognosis is usually good for the A subtype and intermediate for the B. Luminal tumors are the most frequent type of BC, with the A subtype accounting for 40%, and the B subtype for 20% of all patients. HER2 tumors account for 15–20% of patients and lack ESR and PR expression, while overexpressing HER2. They are highly proliferative tumors with intermediate prognosis. Ultimately, triple-negative breast cancer (TNBC), the least common subtype (10–20% of patients), lacks ESR, PR and HER2 expression; it is poorly differentiated and highly proliferative, leading to the worst patient prognosis [16,17,18].

The vast majority of BC-related deaths are not associated with primary tumor (PT) outgrowth. Rather, cancer mortality is generally (>90%) due to metastatic relapse [19,20], which rapidly results in multi-organ failure [21]. It is estimated that 20–30% of early stage BC patients will develop metastatic disease [22], while 5–10% of patients present metastases already at diagnosis [23]. The 5-year survival rate for women with metastatic BC ranges between 18% and 36% [24], compared to >90% of non-metastatic BC patients [25]. Despite the significant therapeutic progresses made in the last few years [13], metastatic BC remains mostly incurable: hence, knowledge around cellular and molecular mechanisms of metastatization and new targeted therapeutic approaches are urgently needed [26].

Traditionally, metastatic progression has been depicted as a late process in which the PT needs to grow to a certain size before releasing cells in the circulation [27]. On the contrary, recent evidence suggests that metastasis spreading can be an extremely early event [28,29], with tumor cells disseminating as early as the pre-malignant phase of tumorigenesis [30,31,32]. Consistently, ~1% of BC patients present metastases in the absence of a clearly identifiable PT [33].

Distant organs to which BC preferentially metastasizes are bones (~70%), lungs (~70%) and liver (~60%, [34]). Recent studies reported that commonly investigated parameters such as age at diagnosis, ethnicity and histological grade are almost never associated with sites of metastasis, whereas the subtype correlates with specific sites of colonization [35]. Indeed, bones represent the most prevalent metastatic site in Luminal A and B patients. Conversely, HER2 BC patients show metastases in both bones and liver at comparable levels, while TNBC metastases are mostly localized in bones and lungs [35,36]. The brain represents the least colonized organ across BC subtypes [34], accounting for ~20% of BC metastases, likely due to the tightness of the blood–brain barrier, which hinders extravasation of BC cells in the brain parenchyma [37]. However, patients with brain metastases generally display the worst prognosis (followed by patients with liver metastases [38]), due to the inefficient delivery of chemotherapeutic drugs to the brain [37].

Several studies investigated PT characteristics that correlate with increased metastasis risk in BC, which have been identified in larger tumor size, increased blood/lymphatic vessel and nerve fiber infiltration, ESR/PR negativity and TP53 overexpression [39,40,41]. However, the genetic and phenotypic determinants that specifically ignite the metastatic process within the PT mass are not yet fully understood.

## 2. The BC Metastatic Progression Is a Multistep Process

The BC metastatic disease can be conceptualized as a multistep process (Figure 1), characterized by a series of consecutive events: (i) epithelial-to-mesenchymal transition (EMT) and local invasion of PT cells in the surrounding tissues; (ii) intravasation and survival of tumor cells in the circulatory or lymphatic system; (iii) extravasation of circulating cells through the vascular endothelium into the parenchyma of distant organs; (iv) seeding and clonal expansion of extravasated cells which originate small colonies, henceforth referred to as “micro-metastases”; (v) micro-metastases adaptation to the foreign microenvironment and formation of clinically detectable lesions. Each of these steps will be further characterized below.

### 2.1. Epithelial-to-Mesenchimal Transition

To leave the PT, cancer cells must first undergo a series of transcriptional modifications that will result in a drastic phenotypical change, known as Epithelial-to-Mesenchymal Transition (EMT). EMT is the critical initial step of the metastatic cascade, which leads to loss of epithelial features, followed by acquisition of migratory and invasive capacities. EMT is a physiological program that occurs during embryo development and, in adults, in processes such as wound healing, tissue regeneration and fibrosis [42,43,44]. EMT induces epithelial cells to lose their polarity, to break down cell-to-cell and cell-to-basal lamina junctions, and to acquire mesenchymal phenotypes, such as a spindle-shape morphology, lack of polarization and cytoskeletal rearrangements, which enable contractility and movement [45]. In the cancer context, epithelial cancer cells undergo EMT in the growing tumor as a consequence of exogenous paracrine signals, such as the Transforming Growth Factor beta (TGFβ) and TGFβ-related cytokines, which activate multiple signaling pathways [46,47,48,49,50,51], including Wnt/β-catenin signaling [52,53,54,55,56,57,58], Notch signaling [59,60,61], interleukins [62,63,64] or environmental conditionings from the “reactive” tumor-associated stroma–i.e., fibroblasts, myofibroblasts, endothelial and immune cells, which activate master transcription factors such as SNAIL [65,66,67,68,69], SLUG [50,70,71,72], TWIST [73,74,75] and ZEB1 [76,77,78,79,80]. In all cases, cells undergo profound transcriptional reprogramming, which leads to the loss of epithelial markers (e.g., E-cadherin [81]), to the acquisition of mesenchymal markers (e.g., N-cadherin [82], fibronectin [83] and vimentin [84,85]), to cytoskeleton reorganization [86,87,88], Extracellular Matrix (ECM)-degradation [83,89,90] and, ultimately, increased migratory capacities. Notably, EMT also favors the generation of Cancer Stem Cells (CSC) [91] and prevents apoptosis and senescence via SNAIL and SLUG-mediated downregulation of p53 [92] and ZEB1-mediated downregulation of p63 and p73 [93]. Moreover, EMT increases resistance to multiple cytotoxic treatments, such as paclitaxel, docetaxel, epirubicin and doxorubicin [94,95], as well as to therapies targeting immune checkpoints (e.g., anti PDL1 and anti-CTL4 [96]). All these events are reversible, following a regulated process known as mesenchymal-to-epithelial transition (MET), which occurs when migratory mesenchymal cells have colonized distant sites and must reacquire epithelial features to infiltrate the new tissue [97].

### 2.2. Intravasation and Circulating Tumor Cells

During BC metastatic progression, mesenchymal-like invasive cancer cells enter the vasculature of either neighboring normal tissues or newly formed vessels within the tumor itself. Lymphatic vessels provide alternative routes for cell distribution to secondary organs. In fact, one of the earliest markers of BC metastatic disease is the presence of micro-metastases in the draining lymph nodes close to the PT site, clinically defined as “sentinel lymph nodes” [98]. Despite their early involvement, lymph nodes may represent temporary “pausing” sites but rarely end points for cancer cells [99], which most frequently seed distant regions via hematogenous dissemination. Circulating Tumor Cells (CTCs) are exposed to a variety of conditions that are potent inducers of a specific apoptotic program known as anoikis [100]. These include the flow shear stress, lack of adhesion signals and intracellular oxidative stress. CTCs are also vulnerable to immune system attacks, exerted in particular by Natural Killer (NK) cells [101]. On the other hand, the EMT phenotype is associated with anoikis resistance [102,103] and CTCs may establish interactions with several cell-types that promote their survival and extravasation. Platelets, for example, form a shield around CTCs that protects them from NK cells [104] and may prevent MET and the resulting loss of migratory/invasive traits [105]. Neutrophils also promote CTC survival via physical entrapment and, similarly to platelets, prevent CTC clearance by NK cells [106]. The balance between pro-apoptotic and pro-survival signals is, however, in favor of the first process, since CTC half-life is estimated to be between 1 and 2.4 h [107]. CTC dissemination and homing to specific organs are strongly influenced by circulatory patterns and structural differences in the capillary wall of each organ. As a consequence, metastatic tropism is considered as a passive process [108].

### 2.3. Extravasation

The mechanical entrapment of cancer cells in the capillary bed of a secondary organ causes CTCs to arrest. As anticipated, vessel configuration strongly contributes to determine the site of cancer cell extravasation. The fenestrated sinusoid capillaries of bone marrow and liver facilitate passive CTC extravasation, accounting for the high incidence of bone and liver BC metastases [34]. Conversely, passage through the endothelial tight junctions of lung capillaries or the blood–brain barrier necessitates to initiate specific “extravasation programs” and complex interactions with other cell types. Active extravasation requires cancer cells to pass through the endothelial wall via a process called Trans-Endothelial Migration (TEM; [109]). TEM is mediated by platelets and components of the innate immune system. Platelets interacting with CTCs trigger TEM by releasing TGFβ or enhancing vasculature wall permeability trough the secretion of adenine nucleotides [110]. Similarly, neutrophils, which are recruited by platelet-derived chemokines, adhere to the vessel wall, provide cancer cells with a physical dock and facilitate their extravasation through the secretion of metalloproteinases [106,110]. Inflammatory monocytes, which may differentiate into metastasis-associated macrophages, are recruited via cytokine CCL2 secreted by cancer cells, facilitating vascular permeability, extravasation and seeding into the host tissue parenchyma [111]. In addition to microenvironmental signals, cancer cells undergo TEM via the expression of autocrine enhancers of cell-motility and mediators of vascular permeability, including epiregulin, VEGF, MMPs, COX2 and ANGPTL4 [112,113]. In particular, Angiopoietin-like 4 (ANGPTL4) expression is induced by stromal TGFβ and it primes BC cell extravasation in the lungs via disruption of vascular integrity and TEM induction [114].

### 2.4. Metastatic Colonization

The development of clinically detectable metastatic lesions represents the final and most complex step in the malignant progression of a tumor. Colonization is thought to be a bottleneck of metastasis, as many cancer cells disseminate, but only 0.01% form metastases [99]. Colonization inefficiency is due to the fact that seeded cancer cells may undergo apoptosis or clearance by NK and cytotoxic T cells. Alternatively, infiltrated cancer cells may enter a quiescent state that is triggered by the intrinsically stressful condition of residing into a foreign microenvironment, which lacks all those familiar ECM constituents, stromal cells, signaling factors and mitogenic cues that had sustained their growth in the PT site [115]. As a consequence, metastatic disease may enter a phase of dormancy, which is sustained by clinical observations. A great number (20–45%) of patients who have been successfully treated for their PT never show a relapse after a long period of latency: these patients may harbor a reservoir of indolent disseminated tumor cells (DTCs) or micro-metastatic clusters in distant organs and they are considered to have asymptomatic minimal residual disease, a condition that may last even for decades [116].

Despite its biological and clinical relevance, little is known about the mechanisms that promote and sustain dormancy in the metastatic context, mostly because of the difficulty to study metastatic latency in patients or experimental models (Table 1). However, it has been demonstrated that members of the TGFβ and BMP family, as well as factors present in the peri-vascular niche (i.e., the microenvironment where the vasculature harboring DTC clusters is embedded in) such as Thrombospondin-1 (TSP-1), play a role in promoting dormancy [116,117]. Successful colonization assumes that DTCs sense and respond to survival and proliferative stimuli, escape immune-surveillance, recruit the necessary supporting stroma and expand until they reach overt-metastasis formation. To do this, DTC clusters must possess at least two pre-requisites: (i) the capacity to seed and maintain a population of CSCs, responsible for initiating metastatic expansion and (ii) the ability to thrive in a hostile microenvironment through a program of organ-specific phenotypic adaptation. Adaptive responses, with regard to BC, will be covered in the following paragraphs.

## 3. BC Intra-Tumor Heterogeneity and Metastasis

BC evolves through the accumulation of oncogenic mutations starting from a genetically normal cell, also known as the “cell-of-origin” [1]. The “cell-of-origin” then undergoes clonal expansion, a process that is accompanied by the acquisition of further genetic and phenotypic traits, thereby generating a state of Intra-Tumor Heterogeneity (ITH; [118]). As a consequence, breast tumors, though clonal in origin, become polyclonal systems [119,120], whereby different clones (i.e., populations of cells that originate from a common ancestor) differ in terms of their genomic and phenotypic profiles [121,122,123].

### 3.1. Genetic Heterogeneity

The METABRIC (Molecular Taxonomy of Breast Cancer International Consortium) study [124,125] investigated the intra-tumor genetic heterogeneity of more than 2000 BC patients. This study reported that the mutations of several cancer-driver genes are present uniquely in a fraction of tumor cells, suggesting that populations of BC cells in the same tumor evolve distinct mutational profiles during in situ progression. Similarly, single-cell DNA analyses on patient biopsies revealed that breast tumors are composed of multiple genetic clones harboring distinct mutational profiles [126,127]. In this regard, different genetic clones are generally confined to distinct areas within the PT, although occasionally single clones can spread across multiple geographical regions in the tumor [128,129]. In line with this, a study on HER2 BC reported that the HER2 gene displays regional heterogeneity in terms of Copy Number Variations (CNVs). Notably, patients carrying highly heterogeneous HER2 amplification within the same mass poorly respond to trastuzumab, a monoclonal antibody to HER2, compared to patients with homogeneous HER2 amplification, suggesting that genetic heterogeneity represents a major challenge for BC therapy [130]. Ultimately, three studies by Aparicio and colleagues demonstrated the presence of several mutations in a small fraction of cells in the whole PT, thus suggesting that such mutations occurred at a later phase of cancer progression [131,132,133].

### 3.2. Transcriptional Heterogeneity

BC displays profound phenotypic ITH, with cells of the same PT adopting different transcriptional and metabolic profiles. Bodenmiller and colleagues investigated the expression of 35 different markers in more than 300 patient-biopsies by mass cytometry [134]. In particular, they evaluated, at single-cell spatial resolution, the expression of proteins involved in specific phenotypes, such as hypoxia response, apoptosis, EMT, proliferation and interaction with ECM. Their analyses revealed that breast PTs are organized in communities of cells, which cluster in separate regions of the tumor and display distinct phenotypes [135].

Recently, single-cell RNA sequencing technology has shed further light on phenotypic ITH. An analysis of multiple murine breast tumor models revealed that cells from the same PT can be extremely different in terms of gene expression profiles, with some cells showing activation of proliferation-related genes (e.g., Ki67), while other cells activate master regulators of EMT (e.g., TWIST1), or either basal (e.g., IGFBP5) or mesenchymal (e.g., vimentin) markers [136,137,138]. Single-cell analysis of the human luminal BC cell line MCF7 revealed that in vitro cultured cells could alternatively display two distinct major transcriptional programs: highly proliferative or dormant-like, with the latter showing upregulation of pathways related to stress response, hypoxia and EMT [138]. Consistently, individual PTs from TNBC patients were reported to consist of both aggressive and highly proliferating cells on one side, and slowly proliferating cells on the other [136,139].

### 3.3. Metabolic Heterogeneity

Single-cell transcriptional analysis of the murine BC genetic model MMTV-PyMT revealed that individual tumors may contain both glycolytic cells and cells that preferentially activate oxidative phosphorylation (OXPHOS) [140]. The switch from an oxidative to a glycolytic metabolism correlates with oxygen availability, since cells in hypoxic regions preferentially rely on glycolysis [141]. Consistently, a recent study on TNBC patient biopsies revealed that hypoxic cells hyperactivate glycolysis, while normoxic cells switch towards OXPHOS [142]. Viable cells in the necrotic core of breast tumors (where oxygen levels are extremely low as a consequence of poor vascularization) exhibit increased glucose uptake to fuel the glycolytic pathway [143]. Ultimately, it has also been reported that metabolism varies in the CSC compartment of breast tumors, with CSCs upregulating mitochondrial proteins, glycolysis and anabolic enzymes with respect to non-stem cancer cells [144,145].

### 3.4. Impacts of ITH on Patient Prognosis and Treatment

ITH represents a hurdle for clinicians, as it might jeopardize patient diagnosis and treatment response [146,147,148]. A high degree of ITH correlates with poor BC outcome and metastatic disease [149,150]. A retrospective study on 75 TNBC patients reported that the degree of heterogeneity in the CNV profile correlates with a higher risk of developing distant metastases and poor prognosis [151]. Likewise, another study quantifying the genetic intra-tumor diversity in patient-specific mutational profiles of more than 900 TCGA (The Cancer Genome Atlas) BC patients showed an inverse correlation between ITH and overall survival [152,153]. Moreover, the analysis of estrogen receptor expression across 970 different breast tumors revealed that patients with the most heterogeneous expression display an increased risk of distant metastases [154]. Thus, the co-existence of heterogeneous populations of cells within the same PT favors distant metastases, suggesting that different clones may develop cooperative interactions [155,156]. The role of clonal cooperativity in BC progression has been investigated since the late 1980s by O’Grady and colleagues, exploiting an in vitro model of rat mammary carcinoma. They showed that individual tumors are composed of both myo-epithelioid (M-cells) and epithelioid (E-cells) cells. These two populations interact through a soluble factor released by M-cells that induces collagenase secretion by E-cells, suggesting that the co-existence of two independent subpopulations is required for the expression of invasive traits [157]. Consistently, a recent study by Polyak and colleagues revealed that the metastatic behavior of certain BC clones may be actively sustained by others. Indeed, the paracrine release of IL-11 and Vascular Endothelial Growth Factor-D (VEGF-D) by a restricted clone in the PT was shown to induce microenvironmental changes (e.g., increased permeability of blood and lymphatic vessels, recruitment of pro-metastatic neutrophils), thus supporting the metastatic progression of other clones [158].

## 4. BC Metastatic Progression Is Not a Genetically Selected Trait

As genetic ITH positively correlates with distant metastasis spreading, it can be hypothesized that metastatic disease is indeed a genetically selected trait, which may depend on the occurrence of metastasis-driver mutations. According to this hypothesis, metastatic cells should share most somatic mutations with the whole tumor and be endowed with a separate subset of mutations capable of driving metastatic progression.

Whole Genome Sequencing (WGS) of 442 paired primary-metastasis samples [159] and Whole Exome Sequencing (WES) of 9 stage IV BC patients [160] showed increased mutational burden in metastatic lesions (i.e., single- and multiple-nucleotide variants, indels and structural variants). In both cases, however, candidate metastasis-driver genes were found at a comparable frequency in PTs and metastases (TP53, PIK3CA, ESR1, GATA3, KMT2C, and the EMT genes SMAD4, TCF7L2 and TCF4; [160]). Bioinformatic analyses of metastasis-specific genes in the former study (24% of all metastasis-associated mutations) revealed a likely “passenger-origin” for these mutations (i.e., mutations that do not confer selective advantages to cancer cells [161]). Likewise, a passenger-origin was hypothesized in the rare metastasis-specific mutations found in two independent studies on BC brain metastases [162,163] and in independent cohorts of BC patients [164,165,166,167]. Interestingly, in other cases metastasis-specific mutations have been interpreted as due to anti-cancer treatments [168]. Other reports, instead, showed that the mutational landscape of metastases and matched PTs mostly overlap [161,162,163,164]. This was also shown at a single-cell level by Navin and colleagues, who investigated the mutational profile of 10 patients affected by invasive BC and showed that invasive cancer cells harbor similar CNVs and an almost identical mutational profile [169]. In conclusion, the high genetic ITH of primary BC samples and their genomic similarity with matched metastatic lesions argue against the existence of selectable pro-metastatic genes and suggest a polyclonal origin of metastases, where clusters of genetically heterogeneous cells are shed into circulation, colonize distant organs and generate a secondary metastatic growth, with results similar to PT [165,170,171].

However, although primary and metastatic BC generally share similar genetic landscapes, several reports have shown relevant differences in mutations when metastases arise years after the PT diagnosis [2,172]. Indeed, a pivotal study by Campbell and colleagues revealed that while in the early phases of cell dissemination PT and metastatic genomic profiles were similar, metastases accumulated independent driver and passenger mutations at later phases [173]. Others reported that ~50% of genomic alterations of metachronous metastases could not be scored in the PT, thereby suggesting an independent mutational evolution of metastatic cells [174,175,176]. Importantly, these studies strongly suggest that the PT genomic profile may not be sufficient to assist the choice of targeting therapies for the metastatic disease.

## 5. Adaptive Responses in BC Metastasis

Emerging evidence suggests that the capacity to metastasize is part of an adaptive response of cancer cells to unfavorable micro-environmental conditions, including hypoxia, scarcity of nutrients, endoplasmic reticulum (ER) stress and chemotherapy (Figure 2; [177,178,179]).

### 5.1. Hypoxia

Hypoxia is a common feature of breast tumors and represents a major threat for cancer cell survival during tumor progression [180]. The deregulated growth of tumor masses progressively increases the distance between cancer cells and capillaries, thereby generating a hypoxic condition that hinders survival and proliferation [41]. Cancer cells respond to hypoxia with the stabilization of Hypoxia-Inducible Factor-1α (HIF-1α), which regulates transcription of several target genes, including glucose transporters, glycolysis enzymes and VEGF [181]. VEGF is secreted by BC cells and stimulates the sprouting of new vessels within the tumor mass, a process referred to as tumor neo-angiogenesis. However, these new vessels are leaky and highly permeable, thus facilitating local intravasation of cancer cells and their spreading in the circulation. Consistently, independent preclinical [182] and clinical studies [183,184,185] demonstrated that hypoxia and increased angiogenesis correlate with metastatic progression and poor patient prognosis.

Moreover, hypoxia was mainly shown to foster EMT in BC through upregulation of SNAIL, ZEB1 and TWIST, which in turn regulate cellular migration, loss of cell-to-cell adhesion, local invasion and stemness traits [186]. In line with this, SHARP1-mediated HIF-1α degradation reduces the expression of HIF-1α target genes, thereby severely impairing BC migration in vitro and metastatic progression in vivo [187]. Ultimately, hypoxic BC cells upregulate ANGPTL4 [181], which disrupts endothelial cell-to-cell junctions in lung capillaries, facilitating lung metastatic colonization [114].

### 5.2. Metabolic Stress

The deregulated growth of primary breast tumors is associated with the exhaustion of the local nutrient microenvironment, which leads to progressive nutrient deprivation, the accumulation of waste products and metabolic stress [123]. A pivotal study on transformed mammary cells revealed that glutamine deprivation strongly fosters the expression of stress-response genes (e.g., ATF4, DDIT3 and XBP1), including inflammatory mediators (e.g., KLF4, CCL2, NF-κB1 and IL20) and it increases the migratory phenotype of tumor cells [188]. In addition, a recent study using a panel of BC cell lines revealed that glutamine deficiency leads to addiction of cancer cells to asparagine and the compensatory upregulation of Asparagine Synthetase (ASNS) [189]. Notably, ASNS upregulation stimulates BC migration in vitro and metastasis spreading in vivo through EMT [190], therefore linking glutamine shortage to metastatic progression. Likewise, glucose deprivation was reported to stimulate oxidative stress in MCF7 BC cells [191], which in turn upregulate metastasis-associated genes, including VEGF and CD44 [192,193]. Ultimately, the accumulation of waste products in the tumor microenvironment leads to local acidification, which promotes metastatic progression. As an example, MCF7 chronically exposed to an acidic microenvironment were shown to acquire an invasive EMT phenotype, characterized by vimentin upregulation and E-cadherin downregulation [194]. Coherently, two studies by Lisanti and colleagues reported that BC cells exposed to the glycolytic-byproduct lactate display significantly higher metastatic potential in vivo, while PT growth remains unaffected [195]. Notably, lactate exposure increases the expression of stemness-related genes (including SP1, MAZ, SREBF1 and PAX4), which are associated with increased risk of developing metastases and poor prognosis [196].

### 5.3. ER Stress

Correct protein folding in the ER is fundamental to guarantee cellular homeostasis and survival. When ER protein folding capacity is hampered, unfolded proteins accumulate, threatening cellular homeostasis. The unfolded protein response (UPR) reprograms gene expression pathways in order to buffer the accumulation of aberrant peptides or to promote cellular apoptosis in case ER stress becomes irreversible [197]. ER stress is caused by several perturbations, including hypoxia, nutrient shortage, oxidative stress, chemotherapy administration and deregulated tumor growth [198,199,200]. ER stress is mediated by three main stress sensors: Inositol-Requiring Protein 1α (IRE1α), Protein Kinase RNA-like ER Kinase (PERK) and Activating Transcription Factor 6 (ATF6), which transduce ER-stress signals to the nucleus via three separate branches [201,202]. The upregulation of IRE1α was reported to booster the migratory phenotype of luminal BC cell lines in vitro, through degradation of several tumor suppressor miRNAs [203]. Consistently, the downregulation of the UPR stress sensor ATF6 significantly reduces BC migration and invasion in vitro [204]. In addition, an analysis of BC patient gene-expression profiles revealed that the overexpression of UPR-mediators Rhomboid Domain-Containing Protein 2 (RHBDD2) and Prion Protein (PRNP) is associated with increased metastatic spreading and poor outcome [205,206,207]. On top of that, the downregulation of UPR genes PERK, ATF4 and LAMP3 was shown to inhibit cellular migration and invasion of BC cells upon hypoxic conditions, linking UPR to the hypoxia-induced BC invasive phenotype [208]. Ultimately, the ER stress mediator Endoplasmic Reticulum Oxidoreductase 1 (ERO1) is crucial for the pro-angiogenic role of HIF-1α upon hypoxia. Indeed, ERO1 deficiency significantly abrogates the secretion of pro-angiogenic factors such as VEGF, IGFBP4 and MMP1, thus inhibiting metastatic progression in vivo [209].

### 5.4. Chemotherapy

Despite enormous advances in BC therapy during the last few years, chemotherapy still represents one the most widely adopted therapeutic options [210,211,212]. However, recent evidence suggests that the administration of chemotherapeutic drugs may result in eliciting a pro-metastatic phenotype [213]. A pioneer work by Gao and colleagues revealed that, upon cyclophosphamide administration, BC cells adopt an EMT-like phenotype characterized by reduced proliferation, resistance to apoptosis, upregulation of drug-metabolizing enzymes and formation of chemoresistant metastases [214]. Ran and colleagues showed that breast tumors acquire a pro-metastatic phenotype upon Paclitaxel administration and that is mediated by Toll-like receptor 4 (TLR4), which promotes the release of inflammatory cytokines, including IL10, IL6 and IL1β, which on their turn stimulate the formation of lymphatic vessels in close proximity to the tumor; this is considered a putative path of metastasis spreading [215]. In another study, Paclitaxel was demonstrated to promote the accumulation of macrophages in the tumor microenvironment, which, in turn, induces expression in cancer cells of the invasive isoform of Mammalian-ENAbled Invasive (MENAINV) protein, an actin binding protein involved in the regulation of cell motility, leading to the intravasation and dissemination of cancer cells [216]. Likewise, Paclitaxel was reported to upregulate the mir-21/CDK5 axis, which activates the expression of EMT markers (vimentin and β-catenin), leading to increased metastasis dissemination to the lungs. Indeed, genetic or pharmacological inhibition of mir-21/CDK5 axis prevented Paclitaxel-induced lung metastases [217]. Carboplatin treatment was also shown to increase BC metastasis. It induces the overexpression of the HIF-1α target Glutathione S-Transferase Omega 1 (GSTO1), which, upon binding to type 1-Ryanodine receptor, promotes Ca^2+^ release from ER and the downstream activation of the PYK2-SRC-STAT3 axis, leading to increased expression of pluripotency genes. Intriguingly, the expression of pluripotency genes fosters the acquisition of a stem-like phenotype, which results in increased metastatic burden in the lungs [218]. Ultimately, two independent studies showed that chemotherapy elicits the release of extracellular vesicles in BC. In particular, De Palma and colleagues reported that Paclitaxel administration induces the release of Annexin A6-enriched vesicles by BC cells. These vesicles promote NF-κB-dependent endothelial cell activation, induction of monocyte-attractant chemokines and monocyte expansion in the lungs, priming the pulmonary niche for metastasis seeding [219]. Concordantly, Doxorubicin administration promotes the release of small extracellular vesicles that are enriched for the glycoprotein Pentraxin-related Protein 3 (PTX3). PTX3 binds P-selectin on the surface of vascular endothelial cells, leading to cell proliferation inhibition, increased expression of matrix metalloproteinases and endothelial cell dysfunction. Therefore, PTX3 causes vascular leakiness in the lungs, thus enhancing the pulmonary colonization of chemotherapy-treated BC cells. Indeed, the inhibition of small extracellular vesicle secretion suppresses chemotherapy-induced metastases [220]. Therefore, albeit fundamental for the treatment of BC, chemotherapy can have detrimental effects, fostering a pro-metastatic phenotype that worsens patient prognosis.

**Table 1 ijms-23-06271-t001:** Experimental Assays Employed to Study Metastases.

In Vitro Models	Mouse Models	Zebrafish Models
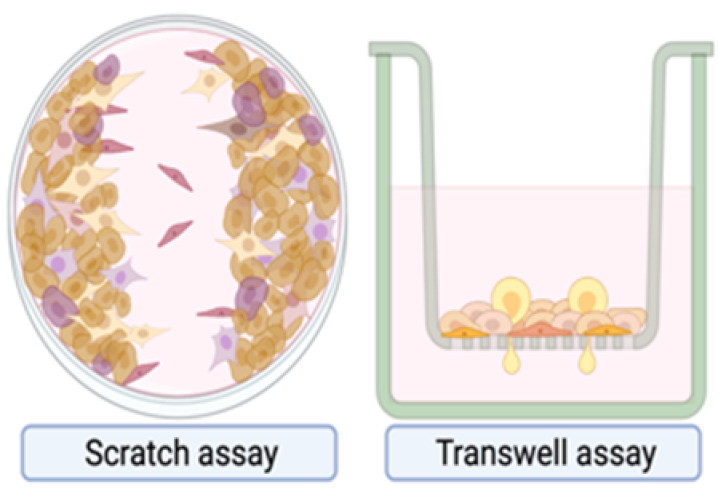	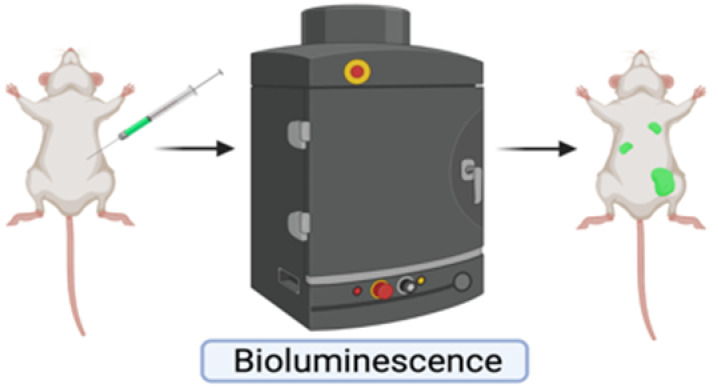	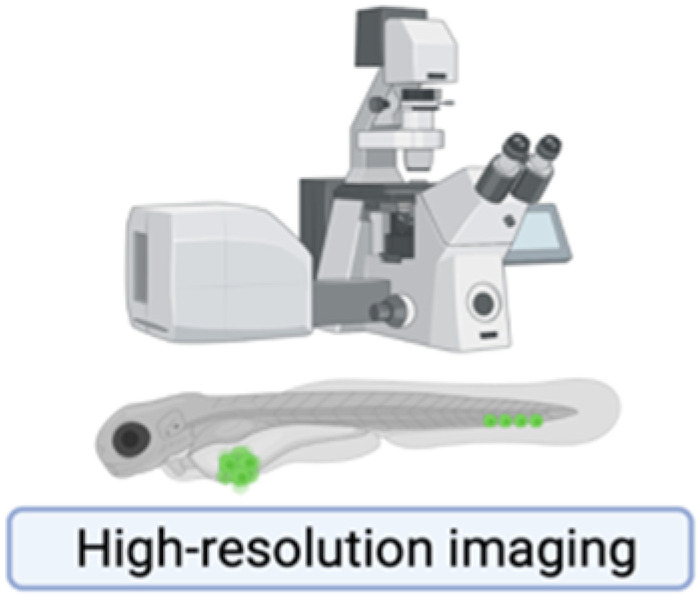
Excellent tools to characterize migration, invasion and adhesion events at molecular level, or for drug testing. Cheap and rapid commercially available platforms. The **scratch assay** exploits a confluent monolayer cell culture in which a linear scratch generates a cell-free area that is replenished by migrating cells. ○ 2D cell migration can be investigated in real-time by time-lapse microscopy [221,222]. ○ Unsuitable for non-adherent cells and for chemotaxis evaluation. The **trans membrane migration assay** (*via* modified Boyden chambers) enables to monitor cell movements between two distinct compartments separated by a microporous membrane. ○ Suitable for chemotaxis evaluation. ○ Suitable for evaluation of cancer cell-ECM interactions by coating the membrane with ECM proteins [223]. ○ Migrating cells can be selectively recovered for further studies. These systems lack a faithful recapitulation of tumor-associated micro-environment and the three-dimensional architecture provided by ECM.	Most appropriate model organisms to investigate human cancer in all its complexity. Genetic engineered mouse models (GEMMs) allow to study the de novo formation of tumors and metastases. ○ They allow for a complete recapitulation of tumor-associated microenvironment. ○ Their drawbacks are inter-individual variability in penetrance and time lagging before metastasis onset [224]. ○ The MMTV-PyMT mouse, obtained through the transgenic expression of Polyomavirus Middle T Antigen, is prone to multifocal mammary carcinomas with 100% penetrance and develop pulmonary metastases in 85% of cases, with a latency of 3 months [225]. Transplantable models can be syngeneic or xenografts. Syngeneic models are obtained by the transplantation of murine cancer cells in mice with matching genetic background. ○ They allow for a complete conservation of the host tumor-associated micro-environment. ○ They may not fully recapitulate human breast cancers. Xenograft models are obtained by the transplantation of human cancer cells into immunocompromised animals. ○ They allow for the recapitulation of human breast cancer features. ○ They do not permit to study interactions with the immune system. Both models can be generated applying two opposite approaches. The experimental metastasis approach is the direct transplantation of cancer cells in the circulation. ○ It ensures rapidity and high reproducibility, by-passing the early steps of the metastatic cascade. ○ It negatively selects dormant pro-metastatic cells. The spontaneous metastasis approach is based on the subcutaneous or orthotopic transplantation of cancer cells in the host. ○ The emergence of distant metastases may be less frequent and highly variable among individuals. ○ It more closely resembles human cancer features, including early steps of the metastatic cascade [224,226]. Imaging metastases in mice often requires euthanasia and post-mortem organ examination. Approaches for live imaging are generally laborious: magnetic resonance imaging, positron enhanced tomography scan and intravital microscopy. Bioluminescence is the simplest live-imaging technique. ○ It relies on detection of photons emitted by genetically-engineered transplanted cancer cells, upon the enzymatic reaction catalyzed by luciferase. ○ Although non-invasive, it has a poor anatomical resolution [227]. Intravital microscopy provides high-resolution and single-cell level visualization of dynamic metastatic events. ○ It exploits surgical optical windows exposed at specific anatomic regions. ○ It provides both spatial and temporal information about cancer cell behavior and enables to follow individual cells over time. ○ It remains experimentally challenging and limited to few specialized laboratories [228].	The use of non-mammalian hosts, as zebrafish, has emerged as an alternative or complementary system to mouse models of cancer metastases [229]. The transparency and small dimensions of zebrafish larvae, together with fluorescently labeled cancer cells, enables high-resolution real-time visualization of: ○ Proliferation, ○ Intravasation, ○ Extravasation, ○ Distant organ colonization by live imaging [230,231,232,233]. The lack of adaptive immune system eliminates the need for immunosuppression. Several transgenic reporter lines with fluorescently labeled components of the host micro-environment (e.g. the vasculature, macrophages and neutrophils) allows for the visualization of complex phenotypes: ○ Neo-angiogenesis, ○ Interaction of human cancer cells with the host innate immune system [234,235,236,237]. Large numbers of animals are attenable, with significantly reduced costs and increased statistical power [238]. These characteristics make the zebrafish xenograft assay an appealing tool which allows to recapitulate and dissect each step of the metastatic cascade in real-time, with an unprecedent rapidity and optical resolution for an in vivo model.

## 6. Concluding Remarks

Metastasis spreading accounts for the vast majority of patient deaths and it represents therefore the deadliest outcome of BC. However, the molecular mechanisms that force cells to abandon the tumor microenvironment and to colonize distant organs are not yet fully understood. In particular, it is not completely clear whether the metastatic phenotype depends on the acquisition of specific metastasis-driver mutations that endow cells with a selective advantage over all the others. In this case, metastasis spreading should represent a genetically selected trait that improves the fitness of specific subpopulations in the PT, by conferring them the capacity to migrate towards distant organs. However, this hypothesis does not properly fit the basic principles of natural selection [239], as metastasizing cells do not display a higher fitness as compared to non-metastasizing ones. Rather, metastasis spreading often represents an inefficient process, in which tumor cells die long before reaching distant organs. On top of that, the outgrowth of BC cells in a different microenvironment may require, even decades after colonization, a period during which PT cells could hugely expand, while the metastatic ones linger in dormancy. Therefore, the hypothesis that metastasis represents a genetically selected trait does not easily fit the Darwinian concepts of selection. In line with this, recent literature largely failed in identifying metastasis-driver mutations (i.e., mutations that characterize the total of metastatic cells and are nearly absent in the PT). This failure can be largely due to the difficulty in having cohorts of patients where PT and metastases are synchronous, as the time-window between PT and metastasis diagnosis comes along with a significant alteration in the mutational profile of metastatic BC cells. This aspect should be carefully considered when studying the mechanisms that underlie metastatization. However, when synchronous primary and distant diseases have been investigated [169,173], results clearly showed that the mutational profile of the two significantly overlap, hence excluding the major role for metastasis-driver mutations in this process. In this review, we focused on this concept, reporting recent evidence that interprete metastatic spreading as an adaptive response to stress conditions (namely, hypoxia, unfolded proteins accumulation, metabolic stress and chemotherapy). Indeed, the important phenotypic determinants of metastatization were identified within BC stress response pathways, whose inactivation turned out to significantly decrease the metastatic progression in preclinical settings. However, the nature and the key players of these adaptive responses are still largely unknown and should be, in our opinion, the major focus of BC metastasis studies in the future (Table 2). In this regard, the use of both *in vitro* and *in vivo* appropriate preclinical models (summarized in Table 1) is of capital importance to dissect the role of specific genes in metastatization and to aggressively determine their exploitability, in order to identify possible drugs which can improve BC patient prognosis in the future.

## Figures and Tables

**Figure 1 ijms-23-06271-f001:**
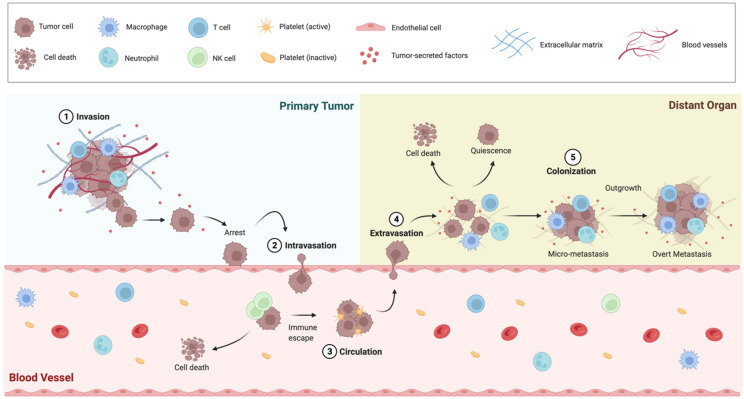
The BC Metastatic Progression is a Multistep Process. The metastatic process implies local invasion of the PT by cancer cells, followed by intravasation in the tumor vasculature. Once arrested in the capillary bed, cells enter the circulatory system. Cancer cells in the circulation are vulnerable to the attacks of the immune system, particularly exerted by Natural Killer cells, which proceed to tumor cell rapid clearance. Immune resistant cancer cells move along the blood vessels as single cells or clusters coated with platelets, and disseminate to secondary sites, passively following the circulatory patterns. Upon their arrival in the capillaries of a distant organ, cancer cells extravasate and start to colonize the foreign parenchyma. Colonization comprises many steps that occur in a timescale of years, during which time cells develop resistance to immunity, adapt to the novel microenvironment and settle in a pre-metastatic niche which support their survival and tumor-initiating capacity. At the metastatic site, cancer cells may be either eliminated or enter in a quiescent state as single cells or micro-metastases. Once the cancer cells break out of dormancy, they reinitiate outgrowth to form an overt metastasis in the distant organ microenvironment (figure created with BioRender.com (accessed on 26 March 2022)).

**Figure 2 ijms-23-06271-f002:**
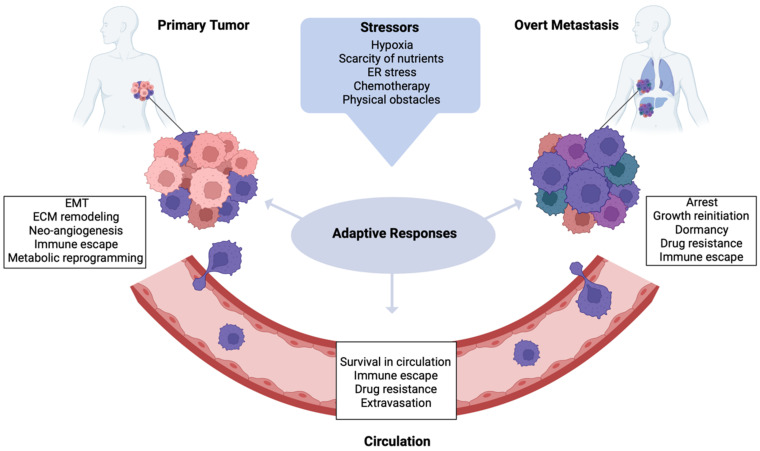
Adaptive Responses in BC Metastatization. During tumor progression, cancer cells encounter different kinds of microenvironmental stressors, such as hypoxia, nutrient deprivation, ER stress and physical obstacles while in transit, besides being exposed to therapeutic drugs. To increase stress tolerance and survive in a hostile environment, cells activate adaptive stress response pathways. These phenotypic adaptations are regulated in a spatial and temporal manner and foster intratumor heterogeneity, thereby endowing a subset of cancer cells with metastatic traits. Adaptive stress responses in the PT lead to EMT, immune escape, metabolic reprogramming and, through active remodeling of ECM and neo-angiogenesis events, enable cells to leave the PT site. Stress signaling also increases the capacity of cancer cells to survive in the circulation and extravasate, eluding immune surveillance and chemotherapy-induced apoptosis. Adaptive pathways at metastatic site regulate the growth dynamics of disseminated cells: once arrested in the target organ, cells can either enter dormancy to tolerate the foreign environment or reinitiate tumor growth (figure created with BioRender.com (accessed on 26 March 2022)).

**Table 2 ijms-23-06271-t002:** Questions to be addressed in future studies on BC metastatization.

1. Despite metastasis is not a genetically selected trait, are there mutational backgrounds that are more prone than others to activate metastasis as an adaptive response to stress?
2. Is the high mutational overlap between primary tumors and metastases due to ecological reasons (i.e., to the necessity of maintaining specific subpopulations at specific frequencies)?
3. Which are the molecular triggers that ignite the passage from micro- to overt metastases?
4. Are mouse models of patient-derived xenografts truly reliable in recapitulating patient’s metastatic progression, since only cancer stem cells survive and form a new tumor upon transplantation?
5. Given the early nature of metastatization, could be worth not to lose more differentiated (“progenitor-like”) cells when modeling the metastatic cascade? In this scenario, could zebrafish be more suitable than mouse in finding “metastasis-prone (differentiated) cells”?

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
