# Peer review of "Beyond Genetics: Metastasis as an Adaptive Response in Breast Cancer"

_ijms, 2022, doi:10.3390/ijms23116271_

Round 1

Reviewer 1 Report

The manuscript of Ruscitto and coworkers illustrates and describe the general principles of the breast cancer metastatic progression, genetic, transcriptional and metabolic heterogeneity of breast tumors, and the mechanisms underlying adaptive responses in breast cancer metastatic progression.

In general, the review is well written and well structured. The bibliography is recent and substantial, although, the work does not make any significant contributions to the breast cancer field. In fact, the manuscript is practically a book chapter. The authors' contribution to the discussion is minimal. In my opinion, for a narrative review the number of authors is too high (excessive).

I don't read this type of review, but in general it is not bad so I will not reject it. However, the authors should specify in the title that the review is "a narrative review", so as not to confuse and/or create false expectations in the reader. Moreover, the authors should add an accurate discussion before the "Concluding Remarks" paragraph, otherwise the work is a mere list of what other researchers have done.

Author Response

We thank the reviewer for having read our manuscript. However, we disagree on several comments. First of all, a quick search on PubMed reveals that 669 works match the query: [(breast cancer[Title]) AND (metastasis[Title]) AND review]. Out of them, 174 were published in the past 2 years, suggesting that there is an urgent need for an update on what is new in the field of breast cancer metastasis. Notably, none of these 174 works investigated in depth whether metastatization depends either on the acquisition of specific mutations or on the activation of adaptive responses to stress. Moreover, we did not find a review paper which clearly compares the role of mutations and phenotypic traits in dictating breast cancer metastatization. With this regard, our work proposed to elucidate the state of the art of this dichotomy: this is the contribution of our work to the breast cancer field. However, we rephrased the “Concluding Remarks” paragraph to accomplish the issue raised by the Reviewer, adding a more in-depth discussion and a section dedicated to open questions in the field.

Secondly, we disagree with the Reviewer regarding the “excessive” number of authors, which is neither too high nor too low. In fact, each author gave his/her proper contribution in writing the manuscript, which would have never been the same without the entire team.

Finally – although we do not completely agree with the Reviewer, since we do not believe to create false expectations with our title – we leave to the Editor the final decision to add the words “narrative review” to the title.

We remain available for further suggestions/comments from the Reviewer, in order to improve the value of our manuscript.

Reviewer 2 Report

In this Review, Ruscitto and colleagues focused on mechanisms of breast cancer metastatization, describing the complex combination of transcriptional and metabolic adaptive events underlying tumor progression. It is really interesting the indication - well documented - that the metastatic processes are not strinctly depending on specific gene mutation or genetic profiles. The review cover all issues.

Author Response

We thank the Reviewer for the appreciation shown for our manuscript. We are grateful for all the positive comments we received.